# ES-RED (Early Seizure Recurrence in the Emergency Department) Calculator: A Triage Tool for Seizure Patients

**DOI:** 10.3390/jcm11133598

**Published:** 2022-06-22

**Authors:** Sung-Eun Lee, Seungyon Koh, Ju-Min Park, Tae-joon Kim, Hee-Won Yang, Ji-Hyun Park, Han-Bit Shin, Bumhee Park, Byung-Gon Kim, Kyoon Huh, Jun-Young Choi

**Affiliations:** 1Department of Emergency Medicine, School of Medicine, Ajou University, Suwon 16499, Korea; plumpboy@hanmail.net (S.-E.L.); 111144@aumc.ac.kr (J.-M.P.); speedheewon@gmail.com (H.-W.Y.); 2Department of Neurology, School of Medicine, Ajou University, Suwon 16499, Korea; esin4498@gmail.com (S.K.); dandy8123@hanmail.net (T.-j.K.); kimbg@ajou.ac.kr (B.-G.K.); khuh@ajou.ac.kr (K.H.); 3Department of Brain Science, School of Medicine, Ajou University, Suwon 16499, Korea; 4Office of Biostatics, Ajou Research Institute for Innovation Medicine, Ajou University Medical Center, Suwon 16499, Korea; jh.park@aumc.ac.kr (J.-H.P.); shin.hanbit@aumc.ac.kr (H.-B.S.); bhpark@aumc.ac.kr (B.P.); 5Department of Biomedical Informatics, School of Medicine, Ajou University, Suwon 16499, Korea; 6Department of Humanities and Social Medicine, School of Medicine, Ajou University, Suwon 16499, Korea

**Keywords:** seizure, recurrence, triage, prediction equation, emergency department

## Abstract

Seizure is a common neurological presentation in patients visiting the emergency department (ED) that requires time for evaluation and observation. Timely decision and disposition standards for seizure patients need to be established to prevent overcrowding in the ED and achieve patients’ safety. Here, we conducted a retrospective cohort study to predict early seizure recurrence in the ED (ES-RED). We randomly assigned 688 patients to the derivation and validation cohorts (2:1 ratio). Prediction equations extracted routine clinical and laboratory information from EDs using logistic regression (Model 1) and machine learning (Model 2) methods. The prediction equations showed good predictive performance, the area under the receiver operating characteristics curve showing 0.808 in Model 1 [95% confidential interval (CI): 0.761–0.853] and 0.805 in Model 2 [95% CI: 0.747–0.857] in the derivation cohort. In the external validation, the models showed strong prediction performance of 0.739 [95% CI: 0.640–0.824] in Model 1 and 0.738 [95% CI: 0.645–0.819] in Model 2. Intriguingly, the lowest quartile group showed no ES-RED after 6 h. The ES-RED calculator, our proposed prediction equation, would provide strong evidence for safe and appropriate disposition of adult resolved seizure patients from EDs, reducing overcrowding and delays and improving patient safety.

## 1. Introduction

Overcrowding and prolonged waiting time in the emergency department (ED) affect the safety and satisfaction of patients, especially critically ill patients. Consequently, efforts have been made to overcome these problems, such as creating a severity triage tool, using a standard working form, and relocating human resources [1,2,3,4,5]. In addition, repeated visits to the ED are one of the overcrowding-causing factors [6]. Therefore, it is essential to establish safe disposition standards for each disease.

Early seizure recurrence in the ED (ES-RED) within 24 h occurs in 13–18% of patients presenting with resolved seizures, with 85% of them occurring within six hours; therefore, a 6–24 h observation period is recommended for patients with seizures visiting a hospital [7,8,9]. An observation time of > 6 h, sometimes > 24 h due to the lack of convincing criteria for disposition of seizure patients, and the subsequent overcrowding in the ED is associated with patient safety [1,2,10]. Therefore, predicting whether and when ES-RED occurs in patients presenting with resolved seizures would enable timely decisions for their proper and safe disposition. This strategy could help relieve overcrowding and prevent delays in the ED.

A comprehensive analysis for predicting ES-RED and guidance for safe disposition of patients are lacking. Previous studies have focused on the individual risk factors of ES-RED as predictors and have demonstrated that several clinical factors such as age, sex, seizure characteristics, and alcohol consumption or laboratory findings such as venous blood gas, glucose levels, and sodium levels are associated with the ES-RED [7,8,11]. However, such research did not provide a pragmatic measure for the prediction of ES-RED. Therefore, this study aimed to propose a model for predicting ES-RED using routinely evaluated basic clinical information, imaging findings, and laboratory findings to facilitate timely decision making for the safe disposition from the ED of adult patients with resolved seizures in the ED.

## 2. Materials and Methods

### 2.1. Study Design and Population

This retrospective observational cohort study analyzed the electronic medical records of adult patients who presented with seizure as a chief complaint at an ED of a tertiary referral medical institution, from 1 March 2016 to 30 June 2019. The inclusion criteria were (1) age 18 years or older and (2) presenting with a seizure. Exclusion criteria were (1) status epilepticus finally diagnosed by epileptologists, (2) seizures occurred more than 24 h before visiting the hospital, (3) refused to be examined and treated in the ED, (4) transferred to other hospitals within four hours, or (5) had a suspected seizure mimic. A *seizure mimic* was diagnosed through detailed history taking, laboratory and electrophysiological studies in the ED, or during follow-up visits at the outpatient clinic of the neurology department, which included convulsive syncope, hyperventilation syndrome, altered mental state induced by drug intoxication, and psychogenic non-epileptogenic seizure.

Basic patient information was collected, such as gender, age, history of medical illnesses (including neurological illnesses), recent alcohol-drinking habits, sleep condition, and routine laboratory test results. When available, brain imaging and electroencephalography results were collected. We defined ES-RED as the seizure recurrence before discharge or within 24 h of visit [7,8,9]. In the case of discharge within 24 h, the patient and caregiver were recommended to revisit the ED if the seizure recurs. An *acute symptomatic cause* was defined as an acute brain insult temporally related to a seizure occurring within seven days, metabolic derangements detected during ED visits, or drug-induced seizures [12,13,14]. We divided the enrolled patients into ‘ES-RED’ and ‘no-ES-RED’ groups. This study was approved by the Institutional Review Board of Ajou University Hospital (AJIRB-MED-MDB-19-467). The requirement for informed consent was waived due to the study’s retrospective nature.

### 2.2. Development of Prediction Models

The enrolled patients (*n* = 688) were randomly assigned to either the derivation or validation cohort (2:1 ratio). To generate a prediction model for ES-RED, baseline demographics, clinical characteristics, seizure characteristics and triggers, vital signs, neurological exam at presentation, and laboratory and imaging findings were analyzed within the derivation cohort. Then, the prediction models that were generated in the derivation cohort were directly applied to the validation cohort to estimate the predictive performance. Two different models were developed in the study: *Model 1* used conventional logistic regression analysis for selecting variables, whereas *Model 2* was based on a machine learning technique, the least absolute shrinkage and selection operator (LASSO).

### 2.3. Statistical Analysis

Variables are expressed as numbers (percentage) and median values (interquartile range (IQR)). Categorical and continuous variables from the ES-RED and no-ES-RED groups in the derivation cohort were compared using the Chi-squared test, Fisher’s exact test, or Mann–Whitney U test [15,16]. The normality of the distribution was assessed using the Shapiro–Wilk test [17,18].

#### 2.3.1. Model 1

Logistic regression analyses were performed within the derivation cohort to predict ES-RED. First, statistically significant variables from univariate logistic regression analyses (*p* < 0.05) were included in the multivariate logistic regression analysis. Then, clinically relevant and statistically feasible variables (*p* < 0.2) were selected again from the multivariable logistic regression analysis to generate the final beta estimates of the regression equation. The beta estimates were calculated using the backward stepwise logistic regression analysis with intercepts.

#### 2.3.2. Model 2

Model 2 was generated within the derivation cohort using the LASSO machine learning technique. The rationale behind using the LASSO technique was to select the variables out of a large number of relevant variables used in our study. Variables that were statistically significant in the univariate analyses were included in the LASSO analysis. The penalty-tuning parameter (lambda) was estimated using ten-fold cross-validation. The optimal lambda was determined within one standard error of the minimal lambda. The variables selected using the optimal lambda were incorporated into the backward stepwise multivariate logistic regression analysis, as in Model 1, to calculate the beta estimates.

In each prediction model, the values of the generated prediction equations were compared between the derivation and validation cohorts. The median values were compared using the Mann–Whitney U test, and variances were compared using Levene’s test. Receiver operating characteristic (ROC) curve analyses were performed within the derivation and validation cohorts. The area under the ROC curve (AUC), sensitivity, specificity, and accuracy were calculated to measure the predictive performances. The values from the generated equations were further divided into quartiles (Q1–Q4), and the association between the quartiles and the rate of ES-RED was analyzed. The association between the quartiles from each prediction model and ES-RED timing was also analyzed using the Kaplan–Meier curve. Statistical analyses were performed using SPSS 25.0 for Windows (SPSS Inc., Chicago, IL, USA) and R version 3.6.3 (R Foundation for Statistical Computing, Vienna, Austria).

## 3. Results

### 3.1. Characteristics of Study Subjects

The selection flow chart of the study population is shown in Appendix A. A total of 841 adult patients presenting with seizures visited the ED. A total of 6 patients were revisited for seizure recurrence within 24 h after early discharge, and they were classified into the recur group. After exclusions, 688 patients included in the study were randomly assigned to the derivation (*n* = 461) and validation (*n* = 227) cohorts. The derivation cohort patients with ES-RED were older than patients with no ES-RED (50 years (IQR, 41–69 years) vs. 41 years (IQR, 25.5–55 years), *p* < 0.001) and had higher systolic (129 mmHg (IQR, 115.75–158.5 mmHg) vs. 125 mmHg (IQR, 110–140.25 mmHg), *p* = 0.046) and diastolic blood pressure (80 mmHg (IQR, 70–96 mmHg) vs. 78 mmHg (IQR, 67.75–88 mmHg), *p* = 0.023). Neurological abnormalities were more frequently observed in the ES-RED group compared to the no-ES-RED group (38.5% vs. 21.2%, *p* = 0.004), and the Glasgow Coma Scale (GCS) scores were lower (15 (IQR, 12.75–15) vs. 15 (IQR, 15–15), *p* = 0.001). Furthermore, ES-RED group patients were more likely to be on two or more anti-seizure medications (ASMs) (30.8% vs. 15.7%, *p* = 0.002), have two or more seizures within 24 h before the ED visit (42.3% vs. 14.1%, *p* < 0.001), and have acute or remote symptomatic causes detected (35.9% vs. 21.9%, *p* = 0.009) than the no-ES-RED group. In laboratory findings, serum glucose (123.5 mg/dL (IQR, 99.75–155.5 mg/dL) vs. 109 mg/dL (IQR, 97–130 mg/dL), *p* = 0.003), lactate (3.0 mmol/L (IQR, 1.97–7.15 mmol/L) vs. 2.51 mmol/L (IQR, 1.6–4.39 mmol/L), *p* = 0.014), erythrocyte sedimentation rate (9.5 mm/h (IQR, 6.0–23 mm/h) vs. 6.5 mm/h (IQR, 2–16 mm/h), *p* = 0.014), and C-reactive protein (0.205 mg/dL (IQR, 0.07–0.5825 mg/dL) vs. 0.09 mg/dL (IQR, 0.03–0.37 mg/dL), *p* = 0.004) levels were higher, and hemoglobin (13.1 g/dL (range, 11.9–14.15 g/dL) vs. 13.8 g/dL (IQR, 12.5–15.0 g/dL), *p* = 0.002), chloride (99 mmol/L (IQR, 96.75–102 mmol/L) vs. 101 mmol/L (IQR, 99–103 mmol/L), *p* < 0.001), and uric acid levels (5.6 mg/dL (IQR, 4.2–7.2 mg/dL) vs. 6.6 mg/dL (IQR, 4.7–9.2 mg/dL), *p* = 0.004) were lower than in the no-ES-RED group. However, structural abnormalities on the images from computed tomography (CT) (56.3% vs. 50.0%, *p* = 0.173) and magnetic resonance imaging (39.8% vs. 37.5%, *p* = 0.952) did not show statistical differences between the two groups. Patients in ES-RED group received more treatment with intravenous benzodiazepine in the ED (80.8% vs. 18.5%, *p* < 0.001; Table 1).

### 3.2. Main Results

#### 3.2.1. Development of Prediction Models in the Derivation Cohort

In the derivation cohort, we determined independent risk factors for ES-RED using univariate and multivariate logistic regression analyses for Model 1 (Table 2). In the univariate logistic regression, age; taking two or more ASMs; two or more seizures within 24 h before the ED visit; initial GCS score; initial SBP; levels of hemoglobin, serum glucose, albumin, uric acid, potassium, chloride, and lactic acid; and presence of acute or remote symptomatic causes of seizures were significantly associated with ES-RED. After incorporating these variables into the multivariate logistic regression, taking two or more ASMs; two or more seizures within 24 h before the ED visit; initial SBP (in mmHg); hemoglobin level (in g/dL); and serum glucose (in mg/dL), uric acid (in mg/dL), potassium (in mmol/L), and lactate levels (in mmol/dL) were finally selected for generating the following prediction equation (Equation (1); Table 3):(1)$$\begin{array}{cc} (0.923\times \mathrm{Taking} & \mathrm{two}\;\mathrm{or}\;\mathrm{more}\;\mathrm{ASM}s^{†})+(1.514\times \mathrm{Two}\;\mathrm{or}\;\mathrm{more}\;\mathrm{seizures}\;\mathrm{within}\;24\;h^{†})\\ & +(0.020\times \mathrm{Systolic}\;\mathrm{blood}\;\mathrm{pressure}\;)-(0.226\times \mathrm{Hemoglobin}\;\mathrm{level})\\ & +(0.004\times \mathrm{Serum}\;\mathrm{glucose}\;\mathrm{level})-(0.100\times (\mathrm{Serum}\;\mathrm{uric}\;\mathrm{acid}\;\mathrm{level})-(0.540\\ & \times \mathrm{Serum}\;\mathrm{potassium}\;\mathrm{level})+(0.149\times \mathrm{Serum}\;\mathrm{lactate}\;\mathrm{level}) \end{array}$$


† Substitute ‘1’ for ‘yes’ and ‘0’ for ‘no’.

The values from Equation (1) ranged from −4.34 to 3.58 in the derivation cohort. The median value and interquartile range were −2.00 (−2.65 to −1.06). The Shapiro–Wilk test in the derivation cohort yielded that the values did not show normal distribution (*p* < 0.001).

For Model 2, the LASSO machine learning technique was used, and lambda was selected within one standard error of the minimal lambda. Age; taking two or more ASMs; two or more seizures within 24 h before the ED visit; initial SBP; GCS score on arrival; and hemoglobin, serum glucose, uric acid, and lactic acid levels were selected and incorporated into the final variables composing the following Equation (2) (Table 3 and Appendix A):(2)$$\begin{array}{ccc} \left(0.007\times \mathrm{Age}\right) & + & \left(0.909\times \mathrm{Taking}\;\mathrm{two}\;\mathrm{or}\;\mathrm{more}\;\mathrm{ASM}s^{†})+(1.422\times \mathrm{Two}\;\mathrm{or}\;\mathrm{more}\;\mathrm{seizures}\;\mathrm{within}\;24\;h^{†}\right)\\ & & +(0.018\times \mathrm{Systolic}\;\mathrm{blood}\;\mathrm{pressure})-(0.049\times \mathrm{GCS}\;\mathrm{score}\;\mathrm{on}\;\mathrm{arrival})\\ & & -(0.210\times \mathrm{Hemoglobin}\;\mathrm{level})+(0.003\times \mathrm{Serum}\;\mathrm{glucose}\;\mathrm{level})\\ & & -(0.094\times \mathrm{Serum}\;\mathrm{uric}\;\mathrm{acid}\;\mathrm{level})+(0.147\times \mathrm{Serum}\;\mathrm{lactate}\;\mathrm{level}) \end{array}$$

† Substitute ‘1’ for ‘yes’ and ‘0’ for ‘no’.

The values from Equation (2) ranged from −2.70 to 4.13 in the derivation cohort. The median value and interquartile range were −0.38 (−1.09 to 0.51). The Shapiro–Wilk test in the derivation cohort also yielded that the values did not show normal distribution (*p* < 0.001).

In the ROC curve analysis, both equations showed good predictive performances^18^ in the derivation cohort. The AUC values were 0.808 (95% confidential interval [CI] [0.761–0.853]) in Equation (1) and 0.805 (95% CI [0.747–0.857]) in Equation (2) (Figure 1a). Sensitivity, specificity, and accuracy were calculated for each equation. In the derivation cohort, Equations (1) and (2) had 77.3% and 76.0% sensitivity, 74.5% and 75.1% specificity, and 75.0% and 75.2% accuracy, respectively (Table 4). The derivation cohort subjects were divided into quartiles according to the equation outputs, and the ES-RED risk in each quartile was analyzed. The frequency of ES-RED was significantly different among quartile groups, with the higher quartile showing a higher ES-RED frequency; ES-RED rates in Q4 were > 40% in both equations (Equation (1): Q1, 0.9%; Q2, 12.1%; Q3, 15.9%; and Q4, 41.1%; Equation (2): Q1, 2.8%; Q2, 10.3%; Q3, 15.0%; and Q4, 42.1%; Figure 1b).

The cumulative incidence—analyzed using the Kaplan–Meier curve—from both equations showed that Q4 was associated with significantly higher ES-RED rates over time (*p* < 0.001; Figure 1c,d). Previous studies reported that most ES-REDs occurred within 6 h in the ED and stays more than 6 h contributed to overcrowding in the ED [11,13]. After that, we focused on ES-RED after 6 h in the ED. Most ES-REDs (89.3%) occurred within 6 h, similar to the previous report. Those who experienced ES-RED after 6 h were predominantly observed (75%) in the fourth quartile.

#### 3.2.2. Validation of Prediction Equations

We applied the prediction equations directly to the validation cohort to estimate the predictive performances. In the validation cohort, there was no statistically significant difference in other variables except for the more focal features (22.5% vs. 12.4%, *p* = 0.003), two or more seizures within 24 h before presentation (26.9% vs. 18.9%, *p* = 0.016), and the slightly higher potassium level (4.1 mmol/L (IQR, 3.8–4.3 mmol/L) vs. 4.0 mmol/L (IQR, 3.73–4.20 mmol/L), *p* = 0.006) than the derivation cohort (Appendix A).

The values from each prediction equation were calculated and compared between the derivation and the validation cohorts. There were no significant differences between the cohorts with regard to the median values and interquartile ranges (−2.00 (−2.65 to −1.06) vs. −1.95 (−2.77 to −1.00), *p* = 0.9333 for Equation (1), −0.38 (−1.09 to 0.51) vs. −0.28 (−1.19 to 0.82), *p* = 0.6179 for Equation (2)) and variances (*p* = 0.5428 for Equation (1); *p* = 0.3944 for Equation (2)). Consequently, the predictive performances of the prediction equations were analyzed. ROC analyses showed acceptable results in both equations with AUC of 0.739 (95% CI [0.640–0.824]) in Equation (1) and 0.738 (95% CI [0.645–0.819]) in Equation (2) (Figure 2a). In the validation cohort, Equations (1) and (2) had 56.4% and 74.3% sensitivity, 85.9% and 70.5% specificity, and 80.2% and 71.9% accuracy (Table 4).

Similarly, ES-RED rates by quartiles were observed with the derivation cohort (Q1, 8.2%; Q2, 8.3%; Q3, 15.1%; and Q4, 44.2% in Equation (1); Q1, 8.8%; Q2, 11.4%; Q3, 16.7%; and Q4, 35.5% in Equation (2); Figure 2b). Most ES-RED occurred within 6 h, which is consistent with the derivation cohort. Interestingly, only the highest quartile (Q4) showed ES-RED in the validation cohort after 6 h (Figure 2c,d).

Subsequently, patients who experienced ES-RED after 6 h were analyzed. In the total cohort (derivation + validation cohorts), 11 patients with ES-RED after 6 h were identified. None of these 11 patients belonged to Q1 in either Equation (1) or (2), while 9 (81.8%) of these patients belonged to Q4 in Equations (1) and (2), thereby suggesting a disposition criterion based on the generated prediction equations.

## 4. Discussion

In this single-center, retrospective cohort study of adult patients presenting with resolved seizures, we generated prediction models after exploring the factors associated with ES-RED. We found that clinical and laboratory parameters can successfully predict ES-RED, thereby developing two prediction models. Between the two equations, due to Equation (1) being simpler, having a slightly better predictive performance, and requiring fewer variables than Equation (2), we propose using Equation (1) as an ES-RED calculator to predict ES-RED.

Just three studies have investigated the risk factors for ES-RED in adult patients with resolved seizures, to the best of our knowledge. These studies reported that alcoholism, history of seizure, age, gender, number of seizures before hospitalization, and levels of pH, bicarbonate, base excess, lactic acid, sodium, and calcium in venous blood tests were associated with early recurrence of seizures [7,8,11]. However, these findings have limited utility in the clinical field because they did not provide a measure to predict ES-RED. Variables in our prediction equations are consistent with previous studies of risk factors for anytime seizure recurrence. Gultekingil et al., reported younger age, taking multiple ASMs, multiple seizure events within 24 h, and abnormal neurological examination or neuroimaging findings regarding the risk factors for seizure recurrence in the pediatric observation unit of the ED [19]. Kim et al., reported the number of seizures, neurological disorders, and an abnormal EEG finding as significant predictors of seizure recurrence after a single seizure [20]. A review by Rizvi et al., reported that older or younger age, female gender, partial seizure, multiple seizure events, remote symptomatic etiology, and abnormal neurological examination were risk factors for seizure recurrence after the first seizure event [21]. The prediction equations presented in the current study include variables presented in previous studies, such as age, multiple seizures before visiting the ED, abnormal GCS score, glucose level, and serum lactate level. On the other hand, our equations included variables not mentioned previously, namely SBP, hemoglobin level, serum potassium level, and uric acid level.

In our study, uric acid showed a negative correlation with ES-RED. Although the exact underlying mechanism is unclear, some studies have investigated uric acid’s role in seizure disorders. Wang et al., showed a U-shaped association between serum uric acid levels and post-stroke epilepsy [22]. Our institute also reported that low uric acid levels help distinguish refractory status epilepticus from responsive status epilepticus patients [23]. The inverse correlation between uric acid and ES-RED in the current and previous studies suggests a potential beneficial effect of uric acid on various seizure disorders. Further research is needed on uric acid’s role in preventing seizures.

Another notable finding in this study is that neuroimaging and electroencephalography findings were not independently associated with ES-RED. A study reported an increase in hospital stay by approximately three hours for the acquisition of electroencephalography and neuroimaging study [24]. The findings in our study suggest that waiting for several hours in the ED to take electroencephalography and neuroimaging tests is unnecessary unless essential.

Our study showed that patients with the lowest quartile (less than −2.65 in Equation (1) and less than −1.09 in Equation (2)) in ES-RED prediction equations had no recurrence after six hours. These findings could help determine the monitoring duration or disposition of seizure patients with low values in our equations. In addition, patients in the highest quartile (more than −1.06 in Equation (1) or more than 0.51 in Equation (2)) comprised > 80% of patients who suffered from ES-RED after six hours. This finding could help provide evidence for early admission of such patients with high values in Equations (1) and (2) to the observation zone for neurological monitoring. Otherwise, information on the risk of seizure recurrence can be given to the patients and caregivers. These results could have a positive impact on reducing overcrowding in the ED.

The present study has limitations. First, our study is a single-center retrospective observational cohort study. However, the strength is that we included more subjects than in previous studies and proposed the predictive equation using routinely evaluated clinical information and laboratory findings in the ED. Second, our study site was a tertiary referral medical center, and patients with minor symptoms may have been transported to other hospitals. Finally, the ES-RED calculator may seem more complicated than the other scoring systems because not all input values are integers. However, most hospitals use computerized systems and can automatically link laboratory values to ES-RED calculators. This automated system would be easier to apply in a real-world clinical situation because the physician would only need to fill out simple clinical information. In addition, it could be applied as a decision-making system by using artificial intelligence through machine learning techniques.

In summary, our study identifies predictive factors for ES-RED and proposes the *ES-RED calculator*, a prediction equation. Overcrowding and delays in the ED are important issues, and seizure is a commonly reported neurologic symptom in the ED, which requires seizure patients to stay in the ED for a long time. Our identified factors and proposed ES-RED calculator could help reduce overcrowding and delay in the ED through early, safe, appropriate, and convincing disposition of adult resolved seizure patients.

## Figures and Tables

**Figure 1 jcm-11-03598-f001:**
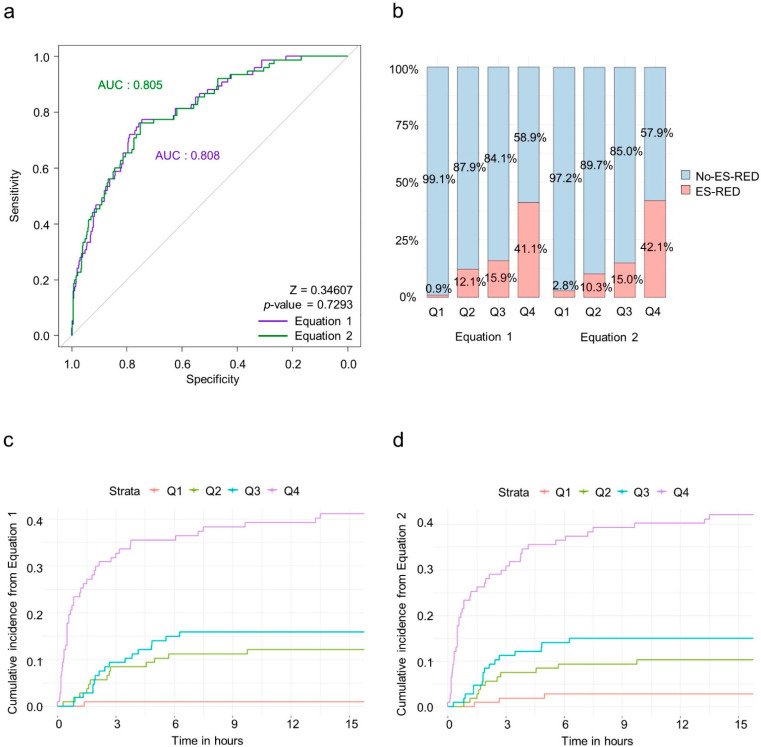
Predictive performance of equations in the derivation cohort. (**a**) Receiver operating characteristic (ROC) curve analysis of both equations. (**b**) Frequency of early seizure recurrence in the emergency department (ES-RED) by quartiles in both Equations. (**c**) Cumulative incidence of ES-RED over time by quartiles in Equation (1). (**d**) Cumulative incidence of ES-RED over time.

**Figure 2 jcm-11-03598-f002:**
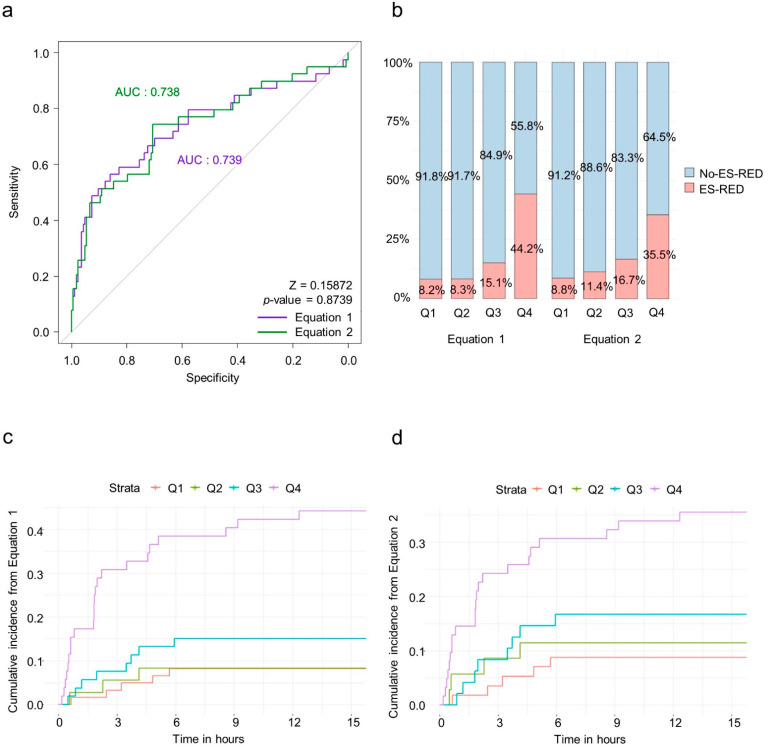
Predictive performance of equations in the validation cohort. (**a**) Receiver operating characteristic (ROC) curve analysis of both equations. (**b**) Frequency of early seizure recurrence in the emergency department (ES-RED) by quartiles in both Equations. (**c**) Cumulative incidence of ES-RED over time by quartiles in Equation (1). (**d**) Cumulative incidence of ES-RED over time.

**Table 1 jcm-11-03598-t001:** General demographics of the derivation cohort.

	No-ES-RED (*n* = 383)	ES-RED (*n* = 78)	*p*-Value
**Demographics**			
Age	41 [25.5–55]	50 [41–69]	<0.001
Sex, female	149 (38.9%)	32 (41.0%)	0.726
**History of medical disease**			0.065
None	266 (69.5%)	43 (55.1%)	
Diabetes/hypertension/dyslipidemia	33 (8.6%)	15 (19.2%)	
Liver	9 (2.3%)	3 (3.8%)	
Kidney	15 (3.9%)	3 (3.8%)	
Thyroid	6 (1.6%)	0 (0.0%)	
Cancer	18 (4.7%)	3 (3.8%)	
Cardiovascular	19 (5.0%)	4 (5.1%)	
Pulmonolgic/rheumatologic/other	17 (4.4%)	7 (9.0%)	
**History of neurological disease**			0.029
None	120 (31.3%)	17 (21.8%)	
Epilepsy	146 (38.1%)	26 (33.3%)	
Stroke	55 (14.4%)	20 (25.6%)	
Brain tumor	9 (2.3%)	5 (6.4%)	
Infection/inflammation	7 (1.8%)	0 (0.0%)	
Other	46 (12.0%)	10 (12.8%)	
**Seizure Characteristics**			
Seizure semiology			0.118
Bilateral impaired awareness motor seizure only	296 (77.3%)	56 (71.8%)	
Focal feature	42 (11.0%)	15 (19.2%)	
Unwitnessed	45 (11.7%)	7 (9.0%)	
Seizure duration			0.156
<3 min	146 (38.1%)	36 (46.2%)	
≥3 min	183 (47.8%)	28 (35.9%)	
Unknown	54 (14.1%)	14 (17.9%)	
Seizure count within 24 h	1 [1–1]	1 [1–3]	<0.001
Seizure count within 24 h ≥ 2	54 (14.1%)	33 (42.3%)	<0.001
Triggering factor			
Alcohol-related	55 (14.4%)	13 (16.7%)	0.601
Sleep deprivation	100 (26.1%)	16 (20.5%)	0.299
Previous seizure history	217 (56.7%)	47 (60.3%)	0.558
Number of prior anti-seizure medication			0.005
None/unknown	241 (62.9%)	37 (47.4%)	
1	82 (21.4%)	17 (21.8%)	
≥2	60 (15.7%)	24 (30.8%)	
Number of prior anti-seizure medication ≥ 2	60 (15.7%)	24 (30.8%)	0.002
**Vital signs and neurological examination**			
Systolic blood pressure, mmHg	125 [110–140.25]	129 [115.75–158.5]	0.046
Diastolic blood pressure, mmHg	78 [67.75–88]	80 [70–96]	0.023
Pulse rate, beats per minute	85 [78–99.25]	90 [80–102]	0.075
Body temperature, °C	36.7 [36.4–36.9]	36.7 [36.475–36.9]	0.487
Glasgow coma score	15 [15–15]	15 [12.75–15]	0.001
**Neurologic examination**			0.004
Normal	302 (78.9%)	48 (61.5%)	
Focal abnormal	21 (5.5%)	6 (7.7%)	
Diffuse abnormal	60 (15.7%)	24 (30.8%)	
**Laboratory findings**			
White blood cell, 10^3^/uL	7.9 [6.2–10.6]	9 [6.5–11.15]	0.302
Red blood cell, 10^6^/uL	4.45 [4.05–4.89]	4.29 [3.91–4.56]	0.002
Hemoglobin, g/dL	13.8 [12.5–15.0]	13.1 [11.9–14.15]	0.002
Mean corpuscular volume, fL	92.4 [89.3–96]	93.6 [89.4–96.8]	0.273
Mean corpuscular hemoglobin, pg	30.9 [29.5–32]	30.9 [29.6–32]	0.933
Mean corpuscular hemoglobin concentration, g/dL	33.3 [32.8–33.8]	33 [32.6–33.6]	0.010
Red cell distribution width, %	13.2 [12.9–13.9]	13.8 [13.2–15]	<0.001
Platelet, 10^3^/uL	225 [183–271]	224 [165.5–270]	0.298
Erythrocyte sedimentation rate, mm/hr	6.5 [2–16]	9.5 [6.0–23]	0.014
C-reactive protein, mg/dL	0.09 [0.03–0.37]	0.205 [0.07–0.5825]	0.004
Glucose, mg/dL	109 [97–130]	123.5 [99.75–155.5]	0.003
Albumin, g/dL	4.5 [4.2–4.8]	4.5 [4.1–4.7]	0.154
Uric acid, mg/dL	6.6 [4.7–9.2]	5.6 [4.2–7.2]	0.004
Creatine kinase, U/L	129 [86–217]	111 [68–223]	0.252
Blood urea nitrogen, mg/dL	11.6 [9.4–14.8]	11.95 [8.8–15.225]	0.931
Creatinine, mg/dL	0.81 [0.69–0.97]	0.82 [0.69–0.97]	0.850
Na, mmol/L	140 [138–141]	139 [136–141]	0.200
K, mmol/L	4.0 [3.8–4.2]	3.9 [3.6–4.1]	0.093
Cl, mmol/L	101 [99–103]	99 [96.75–102]	<0.001
Ca, mg/dL	5.055 [4.6–9.2]	4.98 [4.525–9.075]	0.215
Mg, mg/dL	2.1 [2.0–2.3]	2.1 [1.9–2.2]	0.271
Ammonia, umol/L	28 [19–41]	30 [21–51]	0.213
Lactate, mmol/L	2.51 [1.6–4.39]	3.0 [1.97–7.15]	0.014
pH	7.393 [7.3528–7.4203]	7.383 [7.339–7.424]	0.400
Base Excess, mmol/L	−1.8 [−4.0 to −0.175]	−2.95 [−5.575 to 0]	0.079
Bicarbonate, mmol/L	22.3 [20–24.2]	20.95 [18.85–23.6]	0.097
pCO_2_, mmHg	36.9 [33.3–41.025]	36.55 [32.25–41]	0.865
**Diagnostic evaluation**			
Implemented CT scan			0.173
Normal	184 (48.0%)	29 (37.2%)	
Abnormal	108 (28.2%)	29 (37.2%)	
Not performed	91 (23.8%)	20 (25.6%)	
Implemented MRI scan			0.952
Normal	74 (19.3%)	15 (19.2%)	
Abnormal	49 (12.8%)	9 (11.5%)	
Not performed	260 (67.9%)	54 (69.2%)	
Implemented EEG			0.049
Normal	58 (15.14%)	10 (12.82%)	
Abnormal	87 (22.72%)	28 (35.9%)	
Not performed	238 (62.14%)	40 (51.28%)	
**Etiology**			
Acute symptomatic	20 (5.2%)	8 (10.3%)	0.114
Remote symptomatic	65 (17.0%)	22 (28.2%)	0.021
Any symptomatic	84 (21.9%)	28 (35.9%)	0.009
**IV benzodiazepine in ED**	71 (18.5%)	63 (80.8%)	<0.001

Values are represented as median [interquartile range] or number (percentage). ES-RED, early seizure recurrence in the emergency department; ASM, anti-seizure medication; ESR, erythrocyte sedimentation rate; CRP, C-reactive protein; CT, computed tomography; MRI, magnetic resonance imaging; EEG, electroencephalography; IV, intravenous; ED, emergency department.

**Table 2 jcm-11-03598-t002:** Logistic regression analyses of variables associated with ES-RED in the derivation cohort.

	Univariate Logistic Regression			Multivariate Logistic Regression		
	OR	95% CI	*p*-Value	OR	95% CI	*p*-Value
**Demographics**						
Age	1.027	1.014–1.041	<0.001	1.009	0.991–1.028	0.332
Sex, female	1.093	0.665–1.794	0.727			
**Seizure character**						
Seizure semiology						
Unwitnessed	Reference					
Bilateral impaired awareness motor seizure only	1.216	0.522–2.834	0.650			
Focal feature	2.296	0.852–6.184	0.100			
Seizure duration						
Unknown	Reference					
<3 min	0.951	0.476–1.900	0.887			
≥3 min	0.590	0.290–1.200	0.145			
Seizure count within 24 h	2.362	1.728–3.229	<0.001			
Seizure count within 24 h ≥ 2	4.468	2.621–7.617	<0.001	4.381	2.270–8.455	<0.001
Triggering factor						
Alcohol-related	1.193	0.616–2.309	0.601			
Sleep deprivation	0.730	0.403–1.324	0.301			
Previous seizure history	1.160	0.706–1.905	0.558			
Prior anti-seizure medication ≥ 2	2.393	1.375–4.164	0.002	2.511	1.287–4.900	0.007
**Vital signs and neurological examination**						
Systolic blood pressure, mmHg	1.014	1.004–1.025	0.005	1.018	1.005–1.031	0.007
Diastolic blood pressure, mmHg	1.021	1.006–1.037	0.007			
Pulse rate, beats per minute	1.014	1.000–1.029	0.058			
Body temperature, °C	0.999	0.642–1.554	0.995			
Glasgow coma score	0.856	0.773–0.949	0.003	0.942	0.823–1.078	0.387
**Laboratory findings**						
White blood cell, 10^3^/uL	1.031	0.962–1.105	0.383			
Hemoglobin, g/dL	0.795	0.692–0.914	0.001	0.784	0.648–0.948	0.012
Platelet, 10^3^/uL	0.998	0.995–1.001	0.231			
Erythrocyte sedimentation rate, mm/h	1.013	0.996–1.031	0.143			
C-reactive protein, mg/dL	1.130	0.944–1.352	0.182			
Glucose, mg/dL	1.009	1.004–1.014	<0.001	1.004	0.999–1.009	0.122
Albumin, g/dL	0.632	0.405–0.988	0.044	1.285	0.651–2.534	0.470
Uric acid, mg/dL	0.870	0.793–0.955	0.003	0.916	0.818–1.026	0.131
Creatine kinase, U/L	1.000	0.999–1.001	0.567			
Blood urea nitrogen, mg/dL	1.007	0.982–1.033	0.578			
Creatinine, mg/dL	1.044	0.815–1.337	0.736			
Na, mmol/L	0.967	0.917–1.021	0.227			
K, mmol/L	0.541	0.307–0.952	0.033	0.587	0.292–1.183	0.136
Cl, mmol/L	0.943	0.904–0.984	0.007	0.999	0.947–1.054	0.965
Ca, mg/dL	0.962	0.857–1.080	0.510			
Mg, mg/dL	0.565	0.190–1.685	0.306			
Ammonia, umol/L	1.004	0.996–1.013	0.320			
Lactate, mmol/L	1.137	1.067–1.213	<0.001	1.145	1.049–1.250	0.002
pH	0.095	0.006–1.528	0.097			
Base Excess, mmol/L	0.953	0.903–1.007	0.087			
Bicarbonate, mmol/L	0.956	0.899–1.017	0.153			
pCO_2_, mmHg	1.001	0.973–1.031	0.921			
**Diagnostic evaluation**						
CT finding						
Normal	Reference					
Abnormal	1.704	0.966–3.003	0.065			
Not performed	1.394	0.748–2.599	0.295			
MRI finding						
Normal	Reference					
Abnormal	0.906	0.368–2.233	0.830			
Not performed	1.025	0.547–1.919	0.939			
Abnormal EEG finding	1.867	0.843–4.1328	0.124			
**Etiology**						
Acute symptomatic	2.074	0.879–4.897	0.096			
Remote symptomatic	1.922	1.097–3.367	0.022			
Any symptomatic	1.993	1.183–3.360	0.010	0.916	0.464–1.805	0.799

ES-RED, early seizure recurrence in the emergency department; OR, odds ratio; CI, confidence interval; ASM, anti-seizure medication; SBP, systolic blood pressure; DBP, diastolic blood pressure; PR, pulse rate; BT, body temperature; GCS, Glasgow Coma Scale; Hb, hemoglobin; ESR, erythrocyte sedimentation rate; CRP, C-reactive protein; CT, computed tomography; MRI, magnetic resonance imaging; EEG, electroencephalogram.

**Table 3 jcm-11-03598-t003:** Generation of prediction models in the derivation cohort.

Model 1. Variable Selection Using Logistic Regression Analysis.				
	β	OR	95% CI	*p*-Value
Prior ASMs ≥ 2 (vs. no ASM or 1 ASM)	0.923	2.516	1.293–4.898	0.007
Seizure count within 24 h ≥ 2 (vs. less than 2 seizures)	1.514	4.543	2.415–8.546	<0.001
SBP, mmHg (per 1 mmHg increase)	0.020	1.020	1.007–1.033	0.002
Haemoglobin, g/dL (per 1 g/dL increase)	−0.226	0.797	0.671–0.948	0.010
Glucose, mg/dL (per 1 mg/dL increase)	0.004	1.004	0.999–1.009	0.088
Uric acid, mg/dL (per 1 mg/dL increase)	−0.100	0.905	0.809–1.012	0.080
K, mmol/L (per 1 mmol/L increase)	−0.540	0.583	0.294–1.157	0.123
Lactic acid, mmol/L (per 1 mmol/L increase)	0.149	1.161	1.070–1.259	<0.001
Intercepts	−0.111	0.895		0.954
Equation 1 = (0.923 × Taking two or more ASMs ^†^ ) + (1.514 × Two or more seizures within 24 h ^†^)				
+ (0.020 × Systolic blood pressure) − (0.226 × Haemoglobin level)				
+ (0.004 × Serum glucose level) − (0.100 × Serum uric acid level) − (0.540 × Serum potassium level)				
+ (0.149 × Serum lactate level)				
† Substitute ‘1’ for ‘yes’ and ‘0’ for ‘no’.				
Model 2. Variable selection using LASSO analysis.				
	β	OR	95% CI	*p*-Value
Age (per 1 year increase)	0.007	1.007	0.990–1.025	0.419
Prior ASMs ≥ 2 (vs. no ASM or 1 ASM)	0.909	2.481	1.275–4.828	0.007
Seizure count within 24 h ≥ 2 (vs. less than 2 seizures)	1.422	4.147	2.185–7.872	<0.001
SBP (per 1 mmHg increase)	0.018	1.018	1.005–1.031	0.006
GCS on arrival (per 1 point increase)	−0.049	0.952	0.836–1.084	0.456
Hemoglobin (per 1 g/dL increase)	−0.210	0.811	0.682–0.965	0.018
Glucose (per 1 mg/dL increase)	0.003	1.003	0.998–1.008	0.186
Uric acid (per 1 mg/dL increase)	−0.094	0.911	0.812–1.021	0.110
Lactic acid (per 1 mmol/L increase)	0.147	1.159	1.066–1.260	0.001
Intercepts	−1.767	0.171		0.326
Equation 2 = (0.007 × Age) + (0.909 × Taking two or more ASMs ^†^) + (1.422 × Two or more seizures within 24 h ^†^)				
+ (0.018 × Systolic blood pressure) − (0.049 × GCS score on arrival) − (0.210 × Haemoglobin level)				
+ (0.003 × Serum glucose level) − (0.094 × Serum uric acid level) + (0.147 × Serum lactate level)				
† Substitute ‘1’ for ‘yes’ and ‘0’ for ‘no’.				

OR, odds ratio; CI, confidence interval; ASM, anti-seizure medication; SBP, systolic blood pressure; GCS, Glasgow Coma Scale.

**Table 4 jcm-11-03598-t004:** Predictive performances of prediction equations in the derivation and validation cohorts.

	AUC	95% CI	Sensitivity (%)	Specificity (%)	Accuracy (%)
**Equation (1) (derivation cohort)**	0.808	0.761–0.853	77.3	74.5	75.0
**Equation (1) (validation cohort)**	0.739	0.640–0.824	56.4	85.9	80.2
**Equation (2) (derivation cohort)**	0.805	0.747–0.857	76.0	75.1	75.2
**Equation (2) (validation cohort)**	0.738	0.645–0.819	74.3	70.5	71.9

AUC, area under the receiver operating characteristic curve; CI, confidence interval.

## Data Availability

The data that support the findings of this study are available upon reasonable request to the corresponding author. Direct correspondence regarding this article to Jun Young Choi.

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
