# Peer review of "ES-RED (Early Seizure Recurrence in the Emergency Department) Calculator: A Triage Tool for Seizure Patients"

_jcm, 2022, doi:10.3390/jcm11133598_

Round 1

Reviewer 1 Report

Dear Authors,

I hope my report finds you well and safe. The research approach is appreciated.

The goal of this retrospective study was to propose a model 50 for predicting early seizure recurrence in the emergency department (ES-RED) based on routinely evaluated 51 basic clinical information, imaging findings, and laboratory findings to facilitate timely 52 decision-making for the safe discharge from the ED of adult patients with resolved seizures 53. The authors fulfilled the collecting the data of the patients, performing correlation analysis using logistic regression and machine learning to find out a predictive equation for seizure recurrence and its possible diagnosis. The outcome of the current study provides a new diagnostic tool for seizure recurrence that will help rapidly to shorten the time of diagnosis. Additionally, a receiver operating characteristic curve analysis was performed to discover the sensitivity and specificity of both equations of the suggested model. Therefore, I could conclude that these outcomes answer the research question well and that research provides a new calculator for RED.

Yours

Author Response

Comment: The goal of this retrospective study was to propose a model 50 for predicting early seizure recurrence in the emergency department (ES-RED) based on routinely evaluated basic clinical information, imaging findings, and laboratory findings to facilitate timely decision-making for the safe discharge from the ED of adult patients with resolved seizures. The authors fulfilled the collecting the data of the patients, performing correlation analysis using logistic regression and machine learning to find out a predictive equation for seizure recurrence and its possible diagnosis. The outcome of the current study provides a new diagnostic tool for seizure recurrence that will help rapidly to shorten the time of diagnosis. Additionally, a receiver operating characteristic curve analysis was performed to discover the sensitivity and specificity of both equations of the suggested model. Therefore, I could conclude that these outcomes answer the research question well and that research provides a new calculator for RED.

ANSWER: The authors are grateful for your considerate review.

Reviewer 2 Report

The authors have made a seizure recurrence tool for adults to help improve patient care. I believe they presented valuable data. A few revisions are possible.

1.       In the study design, exclusion criteria included patients with prior seizures more than 24 hours before visiting the hospital. But in the results, table 1 included a line that says “previous seizure history”. I think this should be written more clearly.

2.       There is a paragraph that should not be there. Page 13 Line 351

3.       Are there any more seizure recurrence tools for the emergency department reported in the literature? They should mention such tools in the article. In the introduction section, they may tell how their study differs from previous studies that made seizure recurrence tools. I also think the authors should discuss their article with the previous studies that made such tools. 

Author Response

  1. In the study design, exclusion criteria included patients with prior seizures more than 24 hours before visiting the hospital. But in the results, table 1 included a line that says “previous seizure history”. I think this should be written more clearly.

ANSWER: Thank you for your comment. For the sake of clarity, the exclusion criterion was changed to “seizures occurred more than 24 h before visiting the hospital”. The previous seizure history in the context of a neurological illness of a patient refers to a previous experience of a seizure prior to the event of the ED visit.

  1. There is a paragraph that should not be there. Page 13 Line 351.

ANSWER: We sincerely apologize for the mistake. The relevant change was made in the manuscript.

  1. Are there any more seizure recurrence tools for the emergency department reported in the literature? They should mention such tools in the article. In the introduction section, they may tell how their study differs from previous studies that made seizure recurrence tools. I also think the authors should discuss their article with the previous studies that made such tools.

ANSWER: The authors appreciate this important comment. In fact, only the individual risk factors for ES-RED have been previously investigated, and our attempt is the first to comprehensively analyze and integrate the clinical and laboratory variables to actually predict the facing ES-RED, to triage and guide the patient disposition. Accepting your opinion, the relevant change was made in the introduction section. Findings in previous studies regarding individual predictors of ES-RED are extensively discussed in the discussion section.

Reviewer 3 Report

The abbreviation "early 234 seizure recurrence in the emergency department (ES-RED)" is explained several times in the text. There is no necessity in the harvest, once is enough. Mention of the Mann–Whitney U test and Shapiro-Wilk test requires citation. The presentation of a subject is systematic and comprehensive, list of references is quite full. I am happy to recommend the minor corrections mentioned above.

Author Response

The abbreviation "early 234 seizure recurrence in the emergency department (ES-RED)" is explained several times in the text. There is no necessity in the harvest, once is enough.

ANSWER: Thank you for your comment. Accepting your opinion, relevant changes were made throughout the manuscript. The abbreviation was explained once in the abstract, once in the main manuscript, and in each figure legend.

Mention of the Mann–Whitney U test and Shapiro-Wilk test requires citation.

ANSWER: Citations were added to the corresponding statistical methods. Thank you for the comment.

The presentation of a subject is systematic and comprehensive, list of references is quite full. I am happy to recommend the minor corrections mentioned above.

ANSWER: The authors are grateful for your considerate review.

Reviewer 4 Report

The present work is structured in a linear and coherent manner, with great methodological impact. In this retrospective study, the authors developed two prediction models for the occurrence of epileptic seizures that can be used to reduce crowding and delays in the emergency room and, consequently, improve patient health.

I suggest the following minor revisions:

Minor corrections:

Materials and methods

Table 1: Please enter p-values for all items in each section (e.g., all labels in the History of medical disease section)

Table 2: Please enter p-values for all items in each section (e.g., all labels in the Vital signs and neurological examination section)

Discussion

Line 351-354 : Delete these lines, because they are examples from the JCM template discussion

Author Response

Table 1: Please enter p-values for all items in each section (e.g., all labels in the History of medical disease section)

Table 2: Please enter p-values for all items in each section (e.g., all labels in the Vital signs and neurological examination section)

ANSWER: The authors appreciate your consideration for not missing a trivial detail. In the case of Table 1, we divided the multi-categorized variables into individual categories and compared each variable between ES-RED and no-ES-RED groups. The proportions of each variable were compared using the chi-squared test or Fisher's exact test, according to the sample sizes. [1,2]

Below is the filled-up version of Table 1 (please see the attachment). However, the authors agreed to maintain the original version of Table 1 in the manuscript because the purpose of the analysis was to compare the proportions between the ES-RED group and no-ES-RED group as a whole, and dividing each category would result in the over-fitness of the analysis. In Table 2, the variables in which the p-values were left blank are variables that were not included in the multivariable logistic regression analysis. We are nonetheless grateful for your detailed review.

  1. Nayak, B.K.; Hazra, A. How to choose the right statistical test? Indian J Ophthalmol 2011, 59, 85-86, doi:10.4103/0301-4738.77005.
  2. du Prel, J.B.; Rohrig, B.; Hommel, G.; Blettner, M. Choosing statistical tests: part 12 of a series on evaluation of scientific publications. Dtsch Arztebl Int 2010, 107, 343-348, doi:10.3238/arztebl.2010.0343.

Line 351-354 : Delete these lines, because they are examples from the JCM template discussion

ANSWER: We sincerely apologize for the mistake. The relevant change was made in the manuscript.

Round 2

Reviewer 2 Report

The revisions have been performed satisfyingly.